# Determining Antiradical Capacity of Medicinal Plant Extract Individual Constituents Using Post-Column Reaction Method

**DOI:** 10.3390/ijms25105461

**Published:** 2024-05-17

**Authors:** Jarosław L. Przybył, Jan Stefaniak, Anna Jaroszewicz, Amanda Gawrońska, Marcin Łapiński, Katarzyna Barbara Bączek, Zenon Węglarz

**Affiliations:** Department of Vegetable and Medicinal Plants, Institute of Horticultural Sciences, Warsaw University of Life Sciences, Nowoursynowska 159, 02-776 Warszawa, Polandanna.m.jaroszewicz@gmail.com (A.J.);

**Keywords:** *Saposhnikovia divaricata*, *Astragalus mongholicus*, chromones, coumarins, triterpene saponins, astragalosides, phenolic acids, DPPH, HLPC-DAD

## Abstract

The post-column reaction method enables the evaluation of the antiradical capacity of individual components in a mixture by separating the components using HPLC and measuring stable free radical (e.g., DPPH●) scavenging that occurs after the chromatography column. The equipment typically consists of two detectors. The first records signals of the analytes leaving the column. The second records radical scavenging by the analytes, which appears as a negative band. The recorded signals are found on two separate chromatograms, which must be combined to interpret the results. In this study, a single DAD detector was used behind the post-column reactor, enabling the simultaneous recording of the analyte bands and negative signals, indicating radical scavenging. The objective of this study was to evaluate the antiradical capacity of key compounds found in two herbal raw materials used in traditional Chinese medicine. *Saposhnikovia divaricata* roots contain phenolic acids, chromones, and furanocoumarins. Chlorogenic acid, rosmarinic acid, and imperatorin demonstrated strong radical scavenging, while prim-O-glucoslocimifugin showed a weaker response, both in standards and in root extracts. However, scavenging was not observed for cimifugin and 4′-O-β-D-glucosyl-5-O-methylvisamminol. *Astragalus mongholicus* roots contain astragalosides I-IV (triterpene saponins). None of these showed DPPH● scavenging. Furthermore, additional signals were observed, indicating the presence of unidentified radical scavenging compounds.

## 1. Introduction

Plants are a rich source of valuable compounds, including antioxidants. A number of test methods have been developed to better assess these properties of herbal raw materials and derived products [1]. In recent works, the term ‘antioxidant capacity’ or, more precisely, ‘antiradical capacity’ is used to refer to the determination of antioxidants and their changes over time [2,3,4]. 1,1-diphenyl-2-picrylhydrazyl (DPPH●) is commonly used to evaluate the antioxidant capacity of an extract or solution due to its ability to monitor chemical reactions involving radicals [1,3,5]. DPPH● is a stable free radical that can become a stable molecule by accepting an electron or hydrogen. The results are then compared to a compound with a known antioxidant potential. However, interpreting the results obtained for extracts, which are mixtures with a complex composition, can be challenging and ambiguous [1,4]. Examining the properties of individual components in complex mixtures provides more information, for example, to identify which of the components of a complex mixture are particularly important for the effect of the extract. This is even more important, as the proportions of individual compounds are very specific to individual raw material. The post-column reaction is a promising approach to assessing the antiradical capacity of individual components in a mixture. This is achieved by separating the components of a mixture via HPLC and measuring the scavenging of the stable free radical added to the mobile phase after each of the separated components has left the column. Although this is not a new concept, the method is not widely known or used and, therefore, deserves more recognition [6,7].

*Saposhnikovia divaricata* (Turcz.) Schischk. *Apiaceae* grows wild in northeastern China and Inner Mongolia [8,9]. The herbal raw materials of this species are the roots, which are used as one of the most important drugs in traditional Chinese and Japanese medicine and are officially listed as *Saposhnikoviae Radix* in the pharmacopoeias of China, Japan and Korea, as well as being one of the candidates for inclusion in the German pharmacopoeia (DAB) [9,10,11,12]. The root extracts show antioxidant, antiproliferative, antimicrobial and antiviral activities, as well as anti-inflammatory, antipyretic and anticonvulsant activities [10,13,14]. Two chromones, prim-O-glucosylcimifugin (cimifugin β-D-glucopyranoside) and 4′-O-β-D-glucosyl-5-O-methylvisamminol, which are considered to be mainly responsible for the activity of this raw material, were selected as reference compounds for the identification and quality assessment of *Saposhnikoviae Radix* [12,15]. The raw material also contains coumarins (i.e., cimifugin), furanocoumarins (i.e., imperatorin), phenolics (i.e., chlorogenic acid), polyacetylenes, polysaccharides and a small amount of essential oil [16,17,18,19,20]. Cimifugin and chromones exhibit antipyretic, anti-inflammatory, anti-platelet aggregation and analgesic properties [17,21]. Coumarins and furanocoumarins have antioxidant, antibacterial, antifungal, antiviral, antiallergic, hepatoprotective and anticancer effects, as well as beneficial effects on the cardiovascular and central nervous systems [19,22].

*Astragalus mongholicus* Bunge (syn. *Astragalus membranaceus* (Fisch.) Bge. or *Astragalus membranaceus* (Fisch.) Bge. var. *mongholicus* (Bge.) Hsiao, known as Mongolian milkvetch, *Fabaceae*, is a perennial herbaceous plant found naturally in China, Kazakhstan, Russia, Mongolia and North Korea [12]. The herbal raw material is the roots—*Radix astragali*. It is one of the most important medicinal plants used in traditional Chinese medicine. Triterpene saponins such as astragalosides I-X and isoastragalosides I-IV found in the root demonstrate anti-inflammatory, immunomodulatory, anti-diabetic, cardioprotective, hepatoprotective, anti-aging as well as anti-oxidative activity [23,24,25].

The objective of this study was to evaluate the antiradical capacity of key compounds found in the methanolic extracts of two herbal raw materials used in traditional Chinese medicine. For this purpose, the not new but also not popular post-column reaction method was used. Only specific compounds found in the roots of *S. divaricata* have been shown to scavenge the DPPH radical, indicating antiradical capacity; no compounds characteristic of the roots of *A. mongholicus* have been found to have this effect.

## 2. Results and Discussion

### 2.1. Post-Column Reaction Kit Installation and Commissioning

The high-performance liquid chromatograph (HPLC) is enhanced with an accessory that greatly extends its capabilities. It is used to perform post-column reactions that increase detection capabilities and even allow the detection of analytes that may not have been previously recorded. The attachment also provides the ability to record the scavenging intensity of the stable radical DPPH●. If a radical-scavenging compound is present in the sample, DPPH2 is formed and the sample is bleached, which the diode array UV-VIS (DAD) detector registers as a negative signal at 515 nm, as shown in Figure 1 [26,27]. Some authors have noted that this wavelength may not be appropriate due to the presence of carotenoids, which also absorb light in this range [1]. However, this was not found to be the case. First, methanol (used as solvent) is not a suitable solvent for carotenoids. Second, the chromatographic conditions do not allow for the separation of any possible carotenoids present in the sample.

The apparatus used for post-column reactions usually consists of two detectors [7]. The first one records the signals of the analytes flowing directly from the chromatographic column and the second one records the signals of the upstream analytes. This solution produces a slight shift in the retention times and a change in the bandwidth of the individual analytes, which are recorded on two different chromatograms. These chromatograms should then be superimposed for a better analysis of the results. In the present work, thanks to the use of a single DAD detector, a number of signals were recorded simultaneously, making it possible to obtain chromatograms showing the individual analytes and signals, indicating their possible scavenging of the DPPH● radical at the same time (Figure 2, Figure 3, Figure 4 and Figure 5).

A post-column reactor was inserted between the chromatographic column and the detector, into which the column effluent and the solution supplied by pump C were fed via a T-piece (Figure 2). Once the apparatus was installed, the tightness of the connections was checked and test analyses of the standard mixture were carried out by pumping deionised water with pump C, and the resulting chromatogram was compared with the reference chromatogram obtained without the reactor. As a result of the addition of the module, a slight delay in the retention times of the individual analytes compared with the reference chromatogram, a broadening of the width of the recorded bands at the base (of the order of 2–5%) and a slight increase in system pressure were observed.

### 2.2. Setting of the Operating Parameters

A test mixture of phenolic acid solutions—caffeic acid and rosmarinic acid—was used to select the operating parameters of the system and to check that the radical supplied was scavenged. They have high antiradical capacity, which has been reported several times in the literature [6,28]. The temperature of the reactor was set at 40 °C, the same temperature as the chromatography column. The concentration of the DPPH● solution introduced into the system was then selected. Water–methanol and methanol solutions with concentrations of 3, 5 and 12 mg DPPH● in 100 mL solvent were used. The weighed amount of radical was dissolved in methanol until the solution was clear and then supplemented up to 100 mL with distilled water or methanol. The water–methanol solutions did not give negative signals on the chromatogram at 515 nm, which would indicate the scavenging of the radical by the phenolic acids present in the test mixture. The absence of negative peaks on the chromatogram was considered to be due to an inappropriate solvent and an inappropriate concentration of DPPH●. Subsequent experiments showed that a solution of DPPH● at a concentration of 5 mg × 100 mL^−1^ in 100% methanol gave optimum results. A solution at this concentration had been used previously in similar studies [27]. Negative signals were recorded on the chromatogram, indicating the scavenging of the DPPH● radical by both caffeic and rosmarinic acids. The last parameter to be determined was the flow rate of the DPPH● radical solution. The best results were obtained when a flow rate of 0.5 mL × min^−1^ was used.

### 2.3. Standards Separation

#### 2.3.1. *Saposhnikovia divaricata*

A method has been developed and validated which allows for good separation and the determination of the biologically active compounds characteristic of *S. divaricata* raw material [Appendix A]. The individual compounds differ in their spectra, so a range of analytical wavelengths has been adopted to match them individually (Appendix A). Positive bands (peaks) indicating the presence of investigated compounds were monitored at 218 nm for imperatorin, 230 nm for prim-O-glucosylcimifugin, cimifugin and 4′-O-β-D-glucosyl-5-O-methylvisamminol and at 330 nm for chlorogenic acid and rosmarinic acid. As the roots of this plant contain phenolic acids, which are compounds with known and confirmed antiradical capacity [6,28], they were used to test the method in further studies.

#### 2.3.2. *Astragalus mongholicus*

A mixture of standards of biologically active compounds characteristic of *A. mongholicus* raw material and the two phenolic acids mentioned above was carried out. Astragalosides absorb light very poorly in the UV range, as is well illustrated by the UV-Vis spectra shown in Appendix A. Although it is advisable to exercise caution or abandon the interpretation of the spectrum in the 190–230 nm range due to the highly probable influence of additional factors, the optimum wavelength for monitoring these saponins is 203 nm. However, the choice of such an analytical wavelength may cause a baseline drift phenomenon, especially when a gradient elution is used and a solvent such as acetonitrile is chosen as the mobile phase, as was the case here. Thus, the baseline did not follow a straight line but followed the changing concentration of acetonitrile in the mobile phase during the analysis. Nevertheless, the peaks corresponding to the astraglycosides were sufficiently visible (Appendix A).

The initial analysis of a mixture of solutions containing the four standards (astragalosides I–IV) encountered difficulties in achieving a good separation. Despite the construction of different gradients and the selection of different flow rates, only two bands (peaks) were obtained on the chromatograms. The analysis of each compound separately showed that astragalosides I and II as well as III and IV have such similar retention times that, when a mixture is separated, their signals merge on the chromatogram. Since the main focus of the work was to determine the antiradical capacity of these compounds and not to develop an optimal method for their separation, it was decided to prepare two mixtures of standards—a mixture of astragalosides I and III (Appendix A) and a mixture of astragalosides II and IV (Appendix A). Caffeic acid and rosmarinic acid were added to each as antioxidants with documented activity. Importantly, the phenolic acids were added at an amount 10 times smaller than the astragalosides because they absorb UV light much better than the astragalosides and the aim was to obtain consistent chromatograms.

### 2.4. Determining Antiradical Capacity of Individual Constituents

#### 2.4.1. *Saposhnikovia divaricata*

Once the operating parameters of the whole apparatus had been established and its operation checked, the separation of a mixture of standards of biologically active compounds characteristic of *S. divaricata* was conducted. The results show that chlorogenic acid and rosmarinic acid are effective scavengers of the DPPH● radical (Figure 3). Chlorogenic acid showed a particularly strong scavenging effect. This is in line with the results of the study by Zhang et al. where it was shown that the DPPH● scavenging activity of aqueous and 95% hydroethanolic extracts of the root of *S. divaricata* correlated significantly with the total phenolic content [29]. A large negative band, indicating the scavenging of the DPPH● radical, is also seen for imperatorin, a compound specific to *S. divaricata* species. Patel [22] and Kozioł and Skalicka [19] have reported that coumarins and furanocoumarins exhibit antioxidant effects. Prim-O-glucosylcimifugin shows radical scavenging, but to a lesser extent. Other bioactive compounds, considered as markers for *S. divaricata* species, did not scavenge the DPPH radical. These results may contradict assumptions about the potential antiradical capacity of 4′-O-β-D-glucosyl-5-O-methylvisamminol and cimifugin [10,14], but it should be noted that at least two methods of detecting antiradical capacity are required to fully confirm or exclude these properties [1]. A second reason for the absence of radical scavenging information on the chromatogram may be the insufficient sensitivity of the apparatus or the setting of inadequate parameters to observe such activity.

A study of plant extracts confirms the antiradical capacity of chlorogenic acid and rosmarinic acid (Figure 4). Prim-O-glucosylcimifugin also showed slight scavenging activity. The chromatogram of the plant extract lacks a band corresponding to imperatorin. The rest of the biologically active compounds characteristic of *S. divaricata* again showed no radical scavenging activity. This may confirm that the main antioxidant compound in the plant is chlorogenic acid. An interesting note from the chromatogram of the plant extract is the small negative peak at 4.0 min of the analysis and the rather large negative signal at 5.5 min. The chromatogram showing the separation of the compounds does not show any signals corresponding to the analytes at these minutes. This means that the antiradical capacity is exhibited by an unidentified chemical present in the herbal raw material *S. divaritaca*.

The presence of noise on the baseline was one of the issues that arose during the analyses. The diode array detector (DAD) used in the analyses is sensitive to such distortions caused by uneven pump C operation, which can make it difficult to read the smaller peaks, so the chromatograms showing radical scavenging are compared with reference chromatograms where the compound signals are clearly visible. However, the DAD detector’s ability to record the entire spectrum is offset by its relatively low sensitivity. A contemporary UV-VIS detector that allows for the simultaneous registration of two wavelengths may be a better choice for this type of analysis.

#### 2.4.2. *Astragalus mongholicus*

Apart from the negative signals of caffeic and rosmarinic acids, the chromatograms did not show the expected negative signals corresponding to astragalosides (Figure 5). The absence of a negative signal indicates that the analyte does not scavenge the DPPH● radical, i.e., its antiradical capacity cannot be confirmed. The results obtained from the analysis of a mixture of standards of compounds found in the herbal raw material of *A. mongholicus* thus show that none of the four astragalosides have antiradical capacity. This is in contrast to data in the literature suggesting that astragalosides are good antioxidants [30]. Caffeic acid and rosmarinic acid, which have documented high antiradical capacity [6,28], were added to the mixtures tested to confirm the radical scavenging effect of the method. However, from the presented chromatograms of mixtures containing astragalosides I and III with caffeic and rosmarinic acid (Appendix A) and astragalosides II and IV with caffeic and rosmarinic acid (Appendix A), it is clear that the compounds present in the raw material do not scavenge the radical, i.e., they probably do not have antiradical capacity, whereas the acids present in the mixtures do scavenge the radical, confirming the efficiency of the apparatus. This is all the more significant because, as mentioned above, the mixture contained 10 times fewer acids than astragalosides.

The examination of the plant extract confirms that the compounds in the raw material do not scavenge the DPPH● radical. When analysis was performed without injecting the DPPH● solution into the effluent, no bands (peaks) were recorded on the chromatogram, including signals corresponding to astragalosides I and III as well as II and IV, which overlap. Other unidentified compounds are also present on the chromatogram, but they too do not show radical scavenging. Although pump C, which injects the DPPH● radical, causes waviness and makes the chromatogram difficult to read, a lack of negative signals (DPPH● radical scavenging) was observed at the positions corresponding to the retention times of the astragalosides, which means that the compounds detected in the *A. mongholicus* root extracts are unlikely to have antiradical capacity. It was shown that neither caffeic acid nor rosmarinic acid are present in the *A. mongholicus* raw material. Recent research indicates that the antiradical capacity of these raw material root extracts is attributed to the presence of polysaccharides and flavonoids [31,32,33,34].

## 3. Materials and Methods

### 3.1. Chemicals

Acetonitrile (ACN) gradient grade for liquid chromatography LiChrosolv^®^ Reag. Ph Eur; phosphoric acid suitable for HPLC, LiChropur™, 85%; and 1,1-diphenyl-2-picrylhydrazyl (97%) were purchased from Merck (Poznań, Poland). Water for HPLC was produced in the laboratory using a water purifier that provides high-purity deionised water for laboratory use WCA R03 DP ECO from Cobrabid Aqua (Warsaw, Poland). Standards of chlorogenic acid (3-O-Caffeoylquinic acid), prim-O-glucosylcimifugin (cimifugin β-D-glucopyranoside), rosmarinic acid, cimifugin, 4′-O-β-D-glucosyl-5-O-methylvisamminol, imperatorin (pentosalen), caffeic acid (3,4-Dihydroxycinnamic acid) and astragalosides I-IV were purchased from ChromaDex^®^ (Irvine, CA, USA).

### 3.2. Plant Material

The plants were collected from the Experimental Field of the Department of Vegetable and Medicinal Plants at the Institute of Horticultural Sciences of the Warsaw University of Life Sciences (Warsaw, Poland). Seed material of both species used to establish the experimental field plantation originated from the wild populations in Mongolia. Voucher specimens were deposited in the Herbarium of the Department of Vegetable and Medicinal Plants, Warsaw University of Life Sciences. Roots of three-year-old *S. divaricata* and two-year-old *A. mongholicus* were harvested in late autumn and dried at 60 °C in a convection dryer.

#### 3.2.1. *Saposhnikovia divaricata*

To prepare the root extracts of *S. divaricata*, 1 g of dried raw material was extracted with 100 mL of methanol using the Extraction System B-811 (BÜCHI Labortechnik AG, Flawil, Switzerland). Soxhlet hot extraction was performed using twenty-five extraction cycles, followed by flushing and drying. Extraction parameters were determined based on the results of preliminary tests. The resulting residue was dissolved in 10 mL of methanol. The solution was filtered using a PTFE 0.22 μm pore and 25 mm diameter syringe tip filter (Sigma-Aldrich, Poznan, Poland) and collected in amber glass vials.

#### 3.2.2. *Astragalus mongholicus*

*A. mongholicus* root extract was prepared by flooding 5 g of dried raw material with 2 × 50 mL portions of methanol in a conical flask and subjecting it to a Sonic 6D ultrasonic bath for 20 min at room temperature (Polsonic, Warsawa, Pland). The two resulting extract portions were then combined. The method of extraction and its parameters were chosen based on the results obtained in preliminary tests. The solution was filtered using a PTFE 0.22 μm pore and 25 mm diameter syringe tip filter (Sigma-Aldrich, Poznan, Poland) and collected in amber glass vials.

### 3.3. Post-Column Reaction Kit

A Shimadzu Prominence high performance liquid chromatograph (HPLC) from Shimadzu, Kyoto, Japan was used with an ASI Model 310 Post Column Reactor by Analytical Scientific Instruments, Richmond, USA. The post-column reaction kit consisted of several components, as shown in Figure 2. These included a tank of deionised water acidified with phosphoric acid to pH 3 (1), a tank of acetonitrile (ANC) (2), a degasser DGU-20A3 (3), two LC-20AD pumps as pump A and B (4 and 5), a mixer (6), an auto sampler SIL-20AC HT (7), a chromatographic column Phenomenex^®^ Kinetex^®^ C18, 2.6 μm, 100 Å, 100 mm × 4.6 mm column with a dedicated pre-column (8), a column oven CTO-10AS VP (9), a T-piece (10), an LC-20AT pump C (11), a tank of DPPH● solution in ethanol (12), a post-column reactor ASI model 310 (13), a diode array detector SPD-M20A (14), a system controller CBM-20A (15) and an effluent container (16).

#### 3.3.1. Separation

For *S. divaricata* standards mixture and extracts separation, a binary gradient of mobile phase A (deionised water adjusted to pH 3 with phosphoric acid) and B (ACN) was used as follows: 0.01 min—15% B; 0.50 min—15% B; 4.00 min—70% B; 4.30 min—70% B; 4.50 min—15% B; 7.00 min—stop. The flow rate was set to 1.1 mL × min^−1^, while the oven temperature was maintained at 40 °C. The injection volume used was 5 μL.

For *A. mongholicus* standards mixture and extracts separation, a binary gradient of mobile phase A (deionised water adjusted to pH 3 with phosphoric acid) and B (ACN) was used as follows: 0.01 min—20% B; 1.00 min—20% B; 3.00 min—55% B; 3.50 min—55% B; 4.0 min—20% B; 7.0 min—stop. The flow rate was set to 1.0 mL × min^−1^, while the oven temperature was maintained at 40 °C. The injection volume used was 5 μL.

#### 3.3.2. Post-Column Reaction

The DPPH● solution, with a concentration of 5 mg × 100 mL^−1^, was pumped at a flow rate of 5 mL × min^−1^ and mixed with the column effluent at a T-piece. The resulting mixture was then fed into the post-column reactor, which operated at a pressure of 0.9 MPa and a mixing temperature of 40 °C.

#### 3.3.3. Data Acquisition

The effluent from the reactor was directed towards the diode array detector. Positive bands (peaks) indicating the presence of investigated compounds were monitored for *S. divaricata* standards mixture and extracts at 218 nm for Imperatorin; 230 nm for prim-O-glucosylcimifugin, cimifugin and 4′-O-β-D-glucosyl-5-O-methylvisamminol; and at 330 nm for 3-O-Caffeoylquinic acid (chlorogenic acid). Similarly, for *A. mongholicus* standards mixture and extracts, positive bands (peaks) indicating the presence of astragalosides I-IV were monitored at 203 nm, while for caffeic acid and rosmarinic acid, monitoring occurred at 330 nm. For both species studied, radical scavenging was measured at 515 nm as negative bands (peaks).

## 4. Conclusions

This study represents an initial step to refining a recognized but unpopular post-column reaction method for assessing the antioxidant capacity of plant extract components. A post-column reaction module was connected to a high-performance liquid chromatograph with a single DAD detector, greatly extending its range of applications. In this case, it was possible to assess the antiradical capacity of the individual components of the mixtures, including those that were initially unidentified. The investigation focused on assessing the suitability of a commonly used and easily accessible reagent. The specificity and limitations of this reagent made it possible to verify working conditions, set equipment settings and ensure reliable results. As a result of numerous experiments, suitable parameters for its operation were established, such as the temperature of the reactor, the concentration and the flow rate of the DPPH● radical delivered by pump C. The analysis, storage, processing and interpretation of results are simplified by the modification described.

This study has shown that chlorogenic acid and rosmarinic acid are mainly responsible for the antiradical capacity of *S. divaritaca* root extract. A high potential for DPPH● scavenging by imperatorin has also been demonstrated. A low antiradical capacity can also be seen in the case of prim-O-glucosylcimifugin. The antiradical capacity of cimifugin and 4′-O-β-D-glucosyl-5-O-methylvisamminol cannot be confirmed as they do not exhibit stable free radical scavenging. This opens the way for further research and development of the method using another reagent added after the column. Two negative bands were observed on the chromatogram of the plant extract, indicating the scavenging of the DPPH● radical by the unidentified compounds.

As a result of the analyses carried out, the presence of the biologically active compounds studied in the methanolic extract of *A. mongholicus* root was confirmed. However, none of the four key compounds tested, astragalosides I–IV, showed any antiradical capacity. Only the antiradical capacity of caffeic acid and rosmarinic acid, which were added to the mixture to validate the method, was confirmed. It is possible that measuring the scavenging of this radical is not an adequate way to test the antiradical capacity of these compounds.

Additional signals were observed on the chromatograms, both positive and negative, indicating the presence of unidentified radical scavenging compounds and opening up the field for further research.

## Figures and Tables

**Figure 1 ijms-25-05461-f001:**
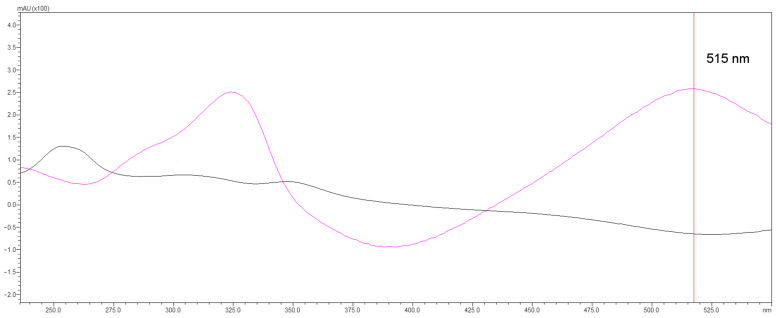
UV-Vis spectra of the DPPH● (purple line) and the DPPH2 (black line). Spectra were recorded as a result of the work carried out in this study.

**Figure 2 ijms-25-05461-f002:**
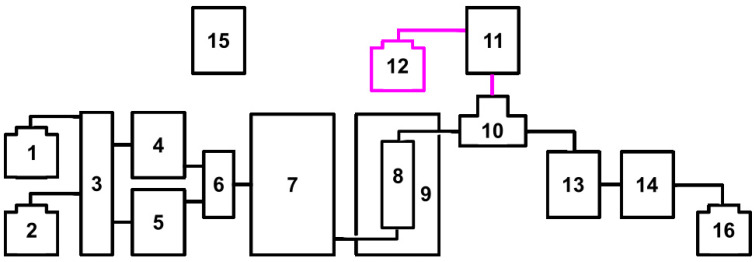
Block diagram of the tested post-column reaction kit. 1—tank of deionised water acidified with phosphoric acid to pH 3; 2—tank of acetonitrile; 3—degasser DGU-20A3; 4—LC-20AD as Pump A; 5—LC-20AD as Pump B; 6—mixer; 7—auto sampler SIL-20AC HT; 8—100 mm × 4.6 mm column Phenomenex^®^ Kinetex^®^ C18, 2.6 μm, 100 Å with pre-column; 9—column oven CTO-10AS VP; 10—T-piece; 11—LC-20AT as Pump C; 12—tank of DPPH● solution in ethanol; 13—post-column reactor ASI model 310; 14—diode array detector SPD-M20A; 15—system controller CBM-20A; 16—effluent container. Purple line: supply of DPPH● solution.

**Figure 3 ijms-25-05461-f003:**
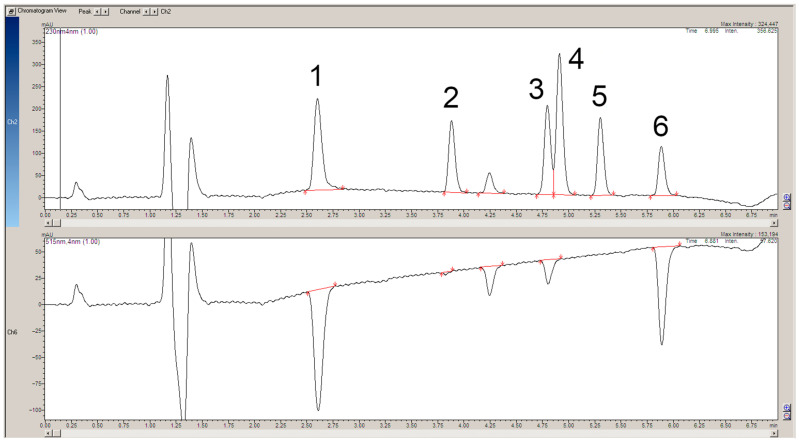
The chromatogram at 230 nm shows a separated mixture of standards that are specific to *S. divaricata* roots (**top**). The 515 nm chromatogram demonstrates the scavenging of the DPPH● radical by the individual separated compounds (**bottom**). Both chromatograms were saved in the same file. 1. Chlorogenic acid (3-O-Caffeoylquinic acid); 2. Prim-O-glucosylcimifugin (Cimifugin β-D-glucopyranoside); 3. Rosmarinic acid; 4. Cimifugin; 5. 4′-O-β-D-glucosyl-5-O-methylvisamminol; 6. Imperatorin (Pentosalen).

**Figure 4 ijms-25-05461-f004:**
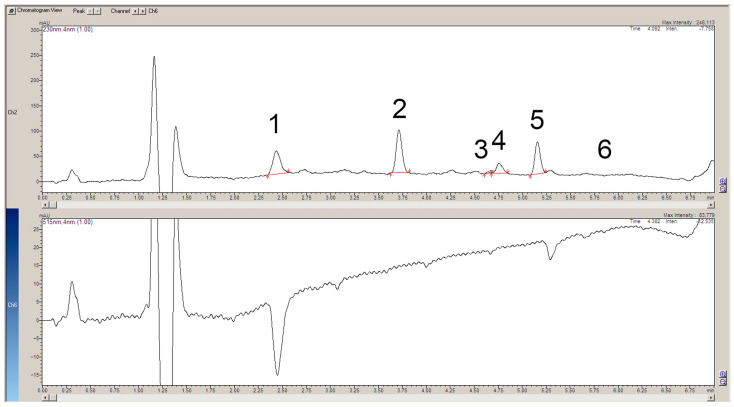
The chromatogram at 230 mn shows the separated compounds present in the methanolic extract of *S. divaricata* roots (**top**). The 515 nm chromatogram demonstrates the scavenging of the DPPH● radical by the individual separated compounds (**bottom**). Both chromatograms were saved in the same file. 1. Chlorogenic acid (3-O-Caffeoylquinic acid); 2. Prim-O-glucosylcimifugin (Cimifugin β-D-glucopyranoside); 3. Rosmarinic acid; 4. Cimifugin; 5. 4′-O-β-D-glucosyl-5-O-methylvisamminol; 6. Imperatorin (Pentosalen) is absent.

**Figure 5 ijms-25-05461-f005:**
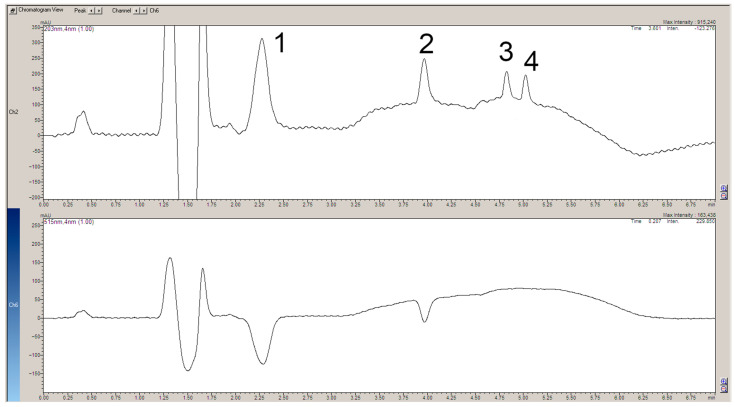
The chromatogram at 203 mn shows the separated compounds present in the methanolic extract of *A. mongholicus* roots along with additionally added caffeic acid and rosmarinic acid (**top**). The 515 nm chromatogram demonstrates the scavenging of the DPPH● radical by the separated compounds (**bottom**). Both chromatograms were saved in the same file. 1. Caffeic acid; 2. Rosmarinic acid. 3. Astragalosides I and III; 4. Astragalosides II and IV.

## Data Availability

The data will be available by contacting the authors.

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
