# Peer review of "Determining Antiradical Capacity of Medicinal Plant Extract Individual Constituents Using Post-Column Reaction Method"

_ijms, 2024, doi:10.3390/ijms25105461_

Round 1
Reviewer 1 Report
Comments and Suggestions for Authors
The manuscript deals with a topic of reasonable interest, and does so with standard appropriate methodology. The rationale, findings and conclusisons are reasonably laid out. A few minor issues:
1. The UV-vis spectra in Figure S1 of the SI should not be shown/interpreted down to 200 nm. Artefacts appear in this region due to water, buffering agent etc. Probably best to only interpret data only from 230-250 nm upwards. The same is true for the Figure S2 spectra. Also, for Figure 1 (presumably; its legend seems incomplete or misplaced). Where the authors do need to still show spectra down to 200 nm (e.g. for the saponin case), proper warning should be given to the reader that the shape of the spectrum is meaningless in this region under these conditions.
2. The notion of "Traditional Chinese Medicine" in the Introduction is too formalized in this expression and risks feeding pseudoscientific discourses on alleged "alternative" kinds of medicine. Perhaps write this without capitalization.
3. The statements on the purported medical effects of these extracts/compounds should be toned down and clarified: most of the cited evidence is indirect, or based on model studies, but not on proper clinical trials. Some of it (as in reference 25) is outright pseudoscientific ("replenishing the qi").
4. In the current form, the manuscript leaves the impression that the authors have just developed/designed this method. While their contribution is important and still uncommon, it is by no means the first. The Introduction and Discussion should be adjusted, to reflect this fact and better mirror the novelty of the study.
Author Response
The manuscript deals with a topic of reasonable interest, and does so with standard appropriate methodology. The rationale, findings and conclusisons are reasonably laid out. A few minor issues:
1. The UV-vis spectra in Figure S1 of the SI should not be shown/interpreted down to 200 nm. Artefacts appear in this region due to water, buffering agent etc. Probably best to only interpret data only from 230-250 nm upwards. The same is true for the Figure S2 spectra. Also, for Figure 1 (presumably; its legend seems incomplete or misplaced). Where the authors do need to still show spectra down to 200 nm (e.g. for the saponin case), proper warning should be given to the reader that the shape of the spectrum is meaningless in this region under these conditions.
- Thank you for drawing attention to this issue. We fully agree with the suggestion and have amended it accordingly. We decided to add the relevant information to all presented spectra due to the specificity of the saponin and the desire to standardise the data presented.
2. The notion of "Traditional Chinese Medicine" in the Introduction is too formalized in this expression and risks feeding pseudoscientific discourses on alleged "alternative" kinds of medicine. Perhaps write this without capitalization.
- Thank you for bringing attention to this aspect. We fully agree with the suggestion and have amended it accordingly.
3. The statements on the purported medical effects of these extracts/compounds should be toned down and clarified: most of the cited evidence is indirect, or based on model studies, but not on proper clinical trials. Some of it (as in reference 25) is outright pseudoscientific ("replenishing the qi").
- Thank you for drawing attention to this issue. We fully agree with the suggestion and have amended it accordingly.
4. In the current form, the manuscript leaves the impression that the authors have just developed/designed this method. While their contribution is important and still uncommon, it is by no means the first. The Introduction and Discussion should be adjusted, to reflect this fact and better mirror the novelty of the study.
- Thank you for bringing this to our attention. We fully agree with the suggestion and have amended it accordingly.
Reviewer 2 Report
Comments and Suggestions for Authors
Dear Authors,
I can hardly find anything to praise you for. You used only one primitive and obsolete method for radical scavenging that does not say much about antioxidant activity. Furthermore, the results of radical scavenging activity are very poor.
It is unclear and in my eyes inconceivable why the authors evaluated and compared radical scavenging of Saposhnikovia divaricata and Astragalus membranaceus var. mongholicus, two different plants belonging to different families containing completely different secondary metabolites. Moreover, it is not clear which plant was used as the authors use the old name of Astragalus mongholicus. Astragalus plants are rich in saponins which we do not expect to have radical scavenging activity, therefore, testing these plants is wasting of time.
Thus, I have no option other than reject your manuscript.
Comments on the Quality of English LanguageModerate editing of English language required.
Author Response
Dear Authors,
I can hardly find anything to praise you for. You used only one primitive and obsolete method for radical scavenging that does not say much about antioxidant activity. Furthermore, the results of radical scavenging activity are very poor.
- Thank you for bringing attention to this aspect of our work. The research presented here represents the initial stage of a broader study on refining and developing post-column reaction methods for testing and evaluating the antioxidant capacity of plant extracts and their components. In this part of the work, we analysed the suitability and feasibility of using a well-known and readily available reagent for this purpose. The specificity and limitations of this reagent enabled us to verify the working conditions, set the equipment parameters, and ensure reliable results.
It is unclear and in my eyes inconceivable why the authors evaluated and compared radical scavenging of Saposhnikovia divaricata and Astragalus membranaceus var. mongholicus, two different plants belonging to different families containing completely different secondary metabolites. Moreover, it is not clear which plant was used as the authors use the old name of Astragalus mongholicus. Astragalus plants are rich in saponins which we do not expect to have radical scavenging activity, therefore, testing these plants is wasting of time.
- Thank you for drawing attention to this aspect of our work. Saposhnikovia was selected as a positive control - a species from which extracts with previously confirmed antioxidant capacity can be obtained. Astragalus was chosen as a negative control to test the developed method.
The results obtained allowed the method to be tested and evaluated, as well as the verification of available data on extracts made from this species.
We agree with the taxonomy suggestion and have made the necessary amendments. Our research has focused on both species for several years.
Thus, I have no option other than reject your manuscript.
- Thank you for expressing your perspective.
Comments on the Quality of English Language
Moderate editing of English language required.
- Thank you for pointing out the issue. We have submitted the manuscript to the proofreading service.
Reviewer 3 Report
Comments and Suggestions for Authors
Figure are presented over the text.
How the color of the vegetal compounds could influence this determination?
Why did you choose to use different ratios between the plant material and the extraction solvent?
The time of reaction between separated compounds and DPPH solution could influence the final results?
Author Response
Figure are presented over the text.
- Thank you for pointing out the flaws. We have made the necessary adjustments.
How the color of the vegetal compounds could influence this determination?
- In the case of the compounds examined in this work, there is no such risk. We describe this in the discussion. However, in the case of pigments such as carotenoids or anthocyanins, it is necessary to verify the lack of influence, which we are currently investigating as part of the method development process.
Why did you choose to use different ratios between the plant material and the extraction solvent?
- Thank you for bringing attention to this aspect. The extraction parameters presented were selected based on the results of the preparatory work carried out at the initial stage of the study. They are not included in this manuscript. We have made adjustments accordingly.
The time of reaction between separated compounds and DPPH solution could influence the final results?
- The reactor used and kit components were configured to minimize the occurrence of such cases.
Reviewer 4 Report
Comments and Suggestions for Authors
Determining Antiradical Capacity of Medicinal Plant Extract Individual Constituents Using Post-Column Reaction Method
I have gone through the entire article, and although I am not an English-language expert, there are issues with sentence design. Authors must pay attention.
In addition, the authors have mentioned the good recovery rate in Table S1, but I am not impressed at all by seeing the plants extract chromatograms. Authors must understand that DAD is not too sensitive.
I am done from my side.
Author Response
Determining Antiradical Capacity of Medicinal Plant Extract Individual Constituents Using Post-Column Reaction Method
I have gone through the entire article, and although I am not an English-language expert, there are issues with sentence design. Authors must pay attention.
- Thank you for pointing out the issue. We have submitted the manuscript to the proofreading service.
In addition, the authors have mentioned the good recovery rate in Table S1, but I am not impressed at all by seeing the plants extract chromatograms. Authors must understand that DAD is not too sensitive.
- Thank you for bringing attention to this aspect. We are fully aware of the limitations of the DAD detector. In this work, however, we have focused on exploiting its unique potential to simultaneously register multiple wavelengths for method development. Indeed, a contemporary UV-VIS detector allowing simultaneous registration of two wavelengths could ultimately be a better choice for this type of analysis. We fully agree with the suggestion and have made adjustments accordingly.
The chromatograms presented show the the great adverse effects of a pump feeding the radical solution, which we describe in detail in the discussion.
I am done from my side.
- Thank you.
Round 2
Reviewer 1 Report
Comments and Suggestions for Authors
The revised version has addressed all my previous concerns, in reasonable manner.
Reviewer 4 Report
Comments and Suggestions for Authors
OK from my side.